# Overview of Research Progress and Application Prospects of Thermal Test Chips

**DOI:** 10.3390/mi16060669

**Published:** 2025-05-31

**Authors:** Lina Ju, Peng Jiang, Yu Ren, Ruiwen Liu, Yanmei Kong, Shichang Yun, Yuxin Ye, Binbin Jiao, Qixing Hao, Honglin Sun

**Affiliations:** 1East China Institute of Photo Electron, Suzhou 215000, China; yuanyuan_514@126.com (L.J.); jp270197195@sina.com (P.J.); 2Institute of Microelectronics of the Chinese Academy of Sciences, Beijing 100029, China; renyu24@ime.ac.cn (Y.R.); liuruiwen@ime.ac.cn (R.L.); kongyanmei@ime.ac.cn (Y.K.); yunshichang@ime.ac.cn (S.Y.); yeyuxin@ime.ac.cn (Y.Y.); 3University of Chinese Academy of Sciences, Beijing 101408, China; 4School of Instrument Science and Technology, Xi’an Jiaotong University, Xi’an 710049, China; 5Suzhou Rich Sensor Technology Co., Ltd., Suzhou 215000, China; hqx_curiosity@outlook.com (Q.H.); sunhonglin2021@126.com (H.S.)

**Keywords:** thermal/stress test, thermal test chip, JESD51-4, microsystem

## Abstract

The development of semiconductor processes and advanced packaging technology has promoted significant advancements in the miniaturization and integration of electronic devices and systems. However, these developments present substantial challenges to the thermal and stress design of current chips, necessitating novel approaches to address these issues. Traditional finite element simulation-assisted design methods have proven inadequate in meeting the demands of highly integrated electronic devices and microsystems due to their inability to effectively simulate the integration process, cross-scale, and multi-physical field coupling. To address these challenges and shorten the design and development period of electronic devices and microsystems, rigorous thermal and stress testing and analyses must be conducted. A promising approach is the utilization of TTC (thermal test chip) technology, a novel in situ testing method, as the primary tool for thermal/stress testing and analyses of the internal interfaces of electronic devices and microsystems. This technology has emerged as a crucial element in validating thermal/stress processes during packaging, as well as in the design of effective heat dissipation solutions. This paper is structured as follows: first, it introduces the principle of thermal test chips; second, it summarizes the domestic and international research progress and index parameter comparison of thermal test chips, as well as the application progress in chip packaging and heat dissipation; and finally, it looks forward to the application prospect of thermal test chips in microsystem design and advanced packaging.

## 1. Introduction

As semiconductor process technology enters the post-Moore era, the development of IC (integrated circuit) chips has shown a variety of trends [1]. The progression of Moore’s law, characterized by the integration of memory-computing capabilities, has led to enhancements in chip performance. However, this advancement has concomitantly resulted in an expansion of chip area and an escalation in power consumption. Conversely, the advent of Moore’s law has catalyzed the emergence of advanced packaging technologies as the pivotal driver of semiconductor evolution [2]. Various advanced packaging technologies, such as SOC (system-on-chip), MCM (multi-chip module), 3D (three-dimensional) stacking, and others, have been employed to enhance the integration of electronic devices and systems. The advent of heterogeneous integration technology has enabled the transition from planar integration to three-dimensional integration, thereby enhancing the functionality of the chip. Additionally, SIP (system-in-package) technology enables the integration of complete microsystems within a single package [3]. Despite the deceleration of Moore’s law, the emergence of new extended Moore technology pathways continues to drive chip development toward miniaturization and higher integration. Consequently, the integration of chips is poised to undergo continuous enhancement.

However, the integration of these components also introduces new challenges, primarily in the form of thermal and stress-related issues [4,5,6]. The most intuitive approach to ensure the reliability of devices is to analyze the temperature distribution of chips and microsystems. Accurate temperature distribution analysis is critical to ensure reliability. Localized hot spots can accelerate electromigration, induce thermo-mechanical stresses, and disrupt material interfaces. For example, in 3D stacked chips, non-uniform temperature distribution may lead to interlayer delamination or bump fracture. Furthermore, temperature gradients directly impact the transistor threshold voltage and carrier mobility, necessitating in situ monitoring to ensure that electrical performance remains within safe operating limits. In the case of large-area chips, which typically utilize FCBGA (flip chip ball grid array) packaging, the issue of heat dissipation is of particular concern due to their substantial power consumption. The increase in chip size, owing to the uneven temperature distribution of stress, as well as the heterogeneity of patches and fillers resulting from thermal mismatch, has led to a significant escalation in the risk of failure due to rupture. In the context of multi-chip high-density integrated microsystems, the internal thermal and stress coupling of the chip assumes greater significance. The chip, during the integration process, introduces process stress and thermal stress, which are induced by the heat generated by deformation and stress concentration. This may result in a series of failure problems, including silicon adapter board fracture, stacked chip fracture, and inter-chip connection bump fracture. Consequently, it is imperative that thermal and stress analyses of chips or microsystems are conducted during the design stage.

Presently, the thermal and stress analyses for the design of electronic chips, devices, and systems are principally based on the combination of finite element simulation and testing [7,8]. The mesh division of the cross-scale, high-precision thermal and stress model is challenging due to the complexity of the division process, which results in extensive calculations. Consequently, the majority of simplified models are employed to address the multi-physical field coupling problem, thereby enhancing the degree of integration. The integration process of electronic devices and microsystems differs due to variations in the simulation of boundary conditions, leading to a reduction in the accuracy of finite element simulations. This underscores the necessity for effective temperature and stress testing methods to assist in the optimization of finite element simulations. The necessity for effective temperature and stress testing methods to assist in optimizing finite element simulations is imperative. Conventional thermal testing methodologies are predominantly indirect in nature, primarily categorized into contact and non-contact modalities. Contact thermal testing requires the pre-positioning of thermocouples, RTDs (resistance temperature detectors), and other temperature sensors within the sample to be tested, which can influence heat transfer of the package structure and the accuracy of the experimental data due to the choice of temperature measurement point. Conversely, non-contact thermal testing employs infrared thermography, thermal reflectance imaging equipment, and analogous methods to measure surface temperature based on the principle of thermal radiation. While this approach can yield high-resolution measurements of the plane thermal distribution, it is limited in its ability to observe and measure internal temperatures within a package chip or microsystem, as it only provides access to the surface temperature. Consequently, the practical applicability of non-contact thermal testing is constrained to a limited range [9,10]. The conventional stress test pathway is comprised of two categories: contact and non-contact. The implementation of a contact stress test necessitates the integration of specific instrumentation, such as strain gauges, optical fibers, and analogous stress-sensing components, directly into or onto the specimen undergoing evaluation. This methodology is susceptible to compromising the stress state of the specimen itself and is constrained in its capacity to generate data from a limited number of test points. This limitation can potentially result in an inadequate dataset if these points fail to encompass the critical stress-affected areas. In contrast, non-contact stress testing predominantly relies on optical methodologies, including scatter interferometry and digital image correlation techniques, among others. While it can produce a higher spatial resolution image of the stress distribution, its application is limited by optical accessibility, and it can only be used to measure stress on the surface of or on partially transparent materials. This makes it difficult to probe the stress inside the package in depth [11,12,13].

A number of domestic and international organizations have proposed using thermal test chips to meet the growing demand for thermal and stress testing of electronic devices and systems. These organizations have conducted research on thermal test chips. This method employs the thermal test chip as a test tool, incorporating heat generation, temperature measurement, and stress measurement units designed on the chip. The chip is then packaged in accordance with the package structure designed for the actual chip, and the heat generation unit can simulate the chip’s heat generation characteristics. The temperature measurement and stress measurement units can realize in situ and accurate testing of temperature and stress at various positions of the chip. Consequently, this method is indispensable in the correction of the accuracy of temperature and stress simulation models, the characterization of the thermal properties of new materials, and the evaluation and optimization of the thermal and stress performance of packaging and heat dissipation. The present paper summarizes the research progress of thermal test chips at home and abroad, as well as the research progress of the application of thermal test chips in advanced packaging, heat dissipation, and microsystems. Finally, it looks forward to the prospect of the development of thermal test chips in the application of microsystem design and packaging. This review synthesizes recent advancements in thermal test chip technology and addresses gaps in understanding multi-physical field coupling in highly integrated microsystems. By systematically comparing domestic and international research advances, this paper provides actionable insights for optimizing thermal and stress design in advanced packages. The exploration of programmable controlled thermal test chips and their applications in transient thermal characterization combines the theoretical framework with practical applications, making it an important reference for engineers and researchers to enhance the reliability and performance of new-generation electronic systems.

## 2. Thermal Test Chip Principles

In multi-chip systems, mutual thermal coupling between components on the same substrate or PCB (printed circuit board) is critical for effective thermal management. For example, neighboring semiconductor chips in a common package transfer heat through shared thermal pathways, the strength of which can be quantified by the transfer thermal resistance (*Rth_,ij_*). This parameter is defined as the temperature rise induced at location i per unit power consumption of source j and reflects the cross-heating effect. As demonstrated in the study of IGBT-diode (insulated gate bipolar transistor-diode) co-packaged devices [14], the transfer thermal resistance can lead to temperature differences in excess of 15 K, underscoring the necessity for accurate modeling to avert reliability degradation. Recent studies have elucidated the pivotal role of mutual thermal coupling in co-packaged devices. For instance, Pawel Górecki et al. [14] determined that IGBTs and anti-parallel diode chips in TO-247 (Transistor Outline 247) packages exhibit temperature variations of up to 20 °C, attributable to the asymmetry between self-heating and transfer thermal resistance. Concurrently, Dirk Schweitzer et al. [15] demonstrated a spatiotemporal dependence of the transfer impedance (*Zth_,ij_*) response in multi-chip modules. These findings underscore the importance of TTIM (transient thermal impedance matrix) models in capturing dynamic cross-coupling effects, particularly in power electronics and heterogeneous integration scenarios. The use of a thermal test chip can provide valuable insights into addressing inter-chip thermal resistance and thermal coupling issues in multi-chip systems.

The thermal test chip is fabricated by arranging multiple independent controllable heaters and temperature and stress measuring elements on the substrate. The heaters are distributed on the surface of the overall chip in a specific layout, which can be used as a stable heat source. The different heaters in the array arrangement can be selected to simulate the heat generation shape and power of the actual chip and electronic devices. Concurrently, the temperature and stress measurement elements can be evaluated in situ to ascertain the temperature and stress distribution on the chip [16]. The standard configuration is illustrated in Figure 1. Thermal test chips are usually processed using standard semiconductor processing techniques, which can be readily combined with a variety of industry-standard packaging and heat dissipation structures. They can also be used in the form of bare chips in packaging, heat dissipation, microsystem integration, and other design studies.

As illustrated in Figure 2, a standard thermal test chip basic unit comprises a heating element, a stress test element, and a temperature measurement element. The heating element is typically a silicon or metal resistor, which is responsible for generating heat to simulate the thermal characteristics of the actual functional chip. To ensure the stability of the thermal characteristics, the heating element is usually required to have low temperature sensitivity. The test elements are utilized to measure the temperature, stress variation, and distribution in situ [17].

In comparison with conventional testing methodologies, the utilization of a thermal test chip as a thermal and stress testing instrument offers several advantages, including simplicity, convenience, detailed data, and high accuracy. Furthermore, the high spatial resolution of the power control and temperature and stress test of the thermal test chip ensures that it can meet the demand for multi-physical field distribution tests of electronic devices and microsystems in complex environments. Furthermore, the thermal test chip’s capacity to operate in both steady state and transient state renders it applicable to a range of applications, including the assessment of transient thermal and stress characteristics of RF (radio frequency) chips, thermal resistance testing, thermal and stress model correction, and the investigation of inter-chip thermal and stress coupling. Moreover, the utilization of thermal test chips to simulate the actual chip thermal characteristics can be synchronized in the early stage of chip development related to package structure testing and optimization. This can result in significant cost savings and a reduction in the chip development cycle.

## 3. Advances in Thermal Test Chip Research

### 3.1. Early Thermal Test Chips

In order to solve the thermal test, it is not possible to obtain the temperature of the device inside the package directly, nor can the package generate a stable and controllable heat source inside the problem. To address this issue, researchers have turned to established semiconductor processing technology to design dedicated thermal test structures for the wafer. These structures are integrated with heating and temperature measurement functions. Utilizing an electrical signal to externally control the wafer enables the generation of a stable and controllable heat flow, facilitating direct temperature reading on the wafer. The thermal test chip is encapsulated using the requisite encapsulation process, with the chip being controlled and read through the encapsulation pins to obtain the direct and accurate junction temperature. This approach ensures the integrity of the encapsulation, as it prevents any additional damage to the encapsulation during the temperature measurement process. In comparison with the conventional physical contact method, the TSP (temperature-sensitive paint) method and the optical method, the TSP method does not result in the destruction of the structure of the package, whilst simultaneously modifying the thermal environment and acquiring a comprehensive two-dimensional temperature distribution [18]. The primary applications of this method include the assessment of thermal resistance in materials, the evaluation of heat dissipation in package structures, and the analysis of the reliability of metal interconnections.

During the 1990s, a number of organizations and companies designed and developed a series of thermal test chips. Séan Cian Ó’Mathuna et al. designed PMOS (P-channel metal oxide semiconductor) and CMOS (complementary metal oxide semiconductor) series of thermal test chips [19,20,21]. The initial thermal test chip, designated as PMOS2, has been fabricated employing the PMOS process, integrating diffusion resistors that function as heating elements and temperature sensors. The chip comprises four substantial diffusion resistors that facilitate regulated heating through the Joule heat effect. The temperature-sensing capability is accomplished through these same diffusion resistors, which are measured by observing the shift in their resistance value in response to temperature variations. Stress monitoring is performed using a four-group aluminum three-wire structure containing both passivated and non-passivated designs to qualitatively analyze mechanical stress, corrosion, and humidity effects through resistance changes caused by metal deformation. The overall chip layout is 9.4 mm^2^ with integrated ring oscillator circuitry and a wire-soldered resistor structure to support package thermal performance evaluation. In the 12 W thermal resistance test, the ceramic PGA (pin grid array) package exhibited a 28% variation in thermal resistance. The calibration temperature range was from 20 °C to 125 °C, and the power density was approximately 127.7 W/cm^2^. However, quantitative accuracy data were not provided for stress monitoring. The chip can be utilized for package thermal resistance testing, humidity dew point detection, and package capacitance characterization. The second thermal test chip, designated as CMOS1, is based on a bimetallic layer CMOS process, with the core heating element comprising a single large-area polysilicon resistor. This resistor possesses a resistance value of 180 Ω, thereby enabling a power density of 48 W/cm^2^ to be realized within a 10 mm^2^ chip. As illustrated in Figure 3, the temperature sensor is implemented through five surface diodes, arranged symmetrically in the central and corner regions of the chip. It employs the linear correlation between diode forward voltage and temperature for temperature measurement, requiring only five signal lines to access all the diodes. The chip design facilitates modular combinations with a base cell size of 2.5 mm^2^ and up to 16 cell structures to simulate the thermal distribution of larger chip sizes. Tests have demonstrated thermal resistance disparities of up to 57% for 24-pin ceramic DIP (dual in-line package) packages. The manufacturing process employs a cross-scribe slot metal interconnect for the validation of thermal management in high-power packages; however, the temperature measurement range remains unspecified.

The third thermal test chip, designated as CMOS2, is a modular design consisting of sixteen 2.5 mm^2^ cells, each equipped with integrated polycrystalline silicon heating resistors and diodes that are galvanically isolated from the n-well. With a total power output of 120 W and an operating voltage of 40 V, the 16 diodes are accessed via eight signal lines to monitor the temperature distribution over the entire chip using the temperature dependence of the forward voltage. The chip features a daisy chain structure, which is employed to assess the mechanical reliability of lead bonding and substrate interconnections within the TAB (tape automated bonding). The calibration temperature range extends from 20 °C to 125 °C, with a statistical error margin of less than ±1% and a systematic error of less than ±10%. The chip was utilized in the ESPRIT-APACHIP (advanced packaging for high performance) program to quantitatively compare MCM thermal techniques, such as heat pipes and immersion cooling, and to verify the effect of cold plate contact pressure on thermal resistance. The fourth thermal test chip, designated as CMOS3, was developed for high-density packaging applications. It utilizes a 600 μm wide aluminum snake resistor as a heating element, with a total power of 60 W, an operating voltage of 5.2 V, and 40 I/O (input/output) ports to ensure uniform current distribution. The chip’s design incorporates 25 diodes in a common anode configuration, a technique that minimizes the number of connecting wires while enabling temperature distribution monitoring. The chip’s layout measures 12 mm^2^, offering 620 I/O ports with a pin pitch of 75 μm, facilitating direct integration into functional substrates. The manufacturing process employs a single-layer metal design, with the exception of the I/O area, thereby reducing development cycles to a single week. The diode calibration slope maintains consistency with CMOS2, exhibiting a slope of −2.6 mV/°C. The chip is employed for the purpose of thermal resistance optimization in 500 W to 5000 W class water-cooled chiller plate systems, wherein a soft metal coating serves to reduce the thermal resistance by 20% to 25%.

James N. Sweet et al. were responsible for the design of the ATC series of thermal test chips [22,23,24] (for example, ATC01 to ATC04 and ATC06, etc.). For instance, in the ATC03 series of chips, a TTC is fabricated using a 1.25 μm CMOS process. The overall chip is square, with a side length of 250 mils (approximately 6.35 mm), as shown in Figure 4. The chip’s architecture comprises four sets of symmetrically distributed polysilicon heating elements situated beneath the aluminum corrosion test structure. These heating elements are energized by a current, thereby generating Joule heat to simulate local or global thermal gradients. The temperature-sensitive sensor element consists of an array of 48 p^+^–n diodes and edge diodes. The array diodes are embedded in a piezoresistive stress sensor cell, and each of these is independently addressable. Temperature measurements are achieved by measuring the diode forward bias voltage (*V_BE_*) as a function of temperature. The linear slope of this measurement is approximately −0.488 °C/mV. The stress sensor element is composed of 48 piezoresistive stress sensor cells, each containing four p-type and four n-type doped resistors arranged in the directions [100] and [110], respectively. Resistor and diode addressing is achieved through CMOS transmission gates, and the change in resistivity of single crystal silicon due to mechanical stress is utilized to invert the stress tensor component. The fabrication process of the chip entails the growth of a 1.9 μm thick field oxide layer on a silicon substrate, the deposition of an aluminum conductor pattern, the implementation of a PETEOS (plasma-enhanced tetraethyl orthosilicate) layer as an interlayer dielectric, the covering of the passivation layer with bonding windows, and the assembly of the chip with a silver-filled thermoplastic binder to ensure its stability. The heater power density is up to 25 W/heater with a total power limit of 100 W. The temperature measurement range is from 25 °C to 100 °C, with practical applications up to 150 °C. The temperature measurement accuracy of the array diode is ±0.006 °C/mV for the slope error, and the edge diode has a slope of −0.761 °C/mV with a standard error of ±0.042 °C/mV. The stress measurement range encompasses both package and thermal stresses. The temperature difference between the experimental and FEM (finite element method) analysis is approximately 3 °C/W, which primarily originates from the thermal resistance of the chip mounting layer. The ATC03 chip is extensively utilized in the measurement of thermal resistance distribution of three-dimensional multi-chip modules, evaluation of the packaging process, reliability testing, and optimization verification of high thermal conductivity materials.

In the early research on thermal test chips, the design of the chip was primarily focused on the design of the basic structure of the thermal test chip, as well as the generation of a stable heat source, the accurate measurement of temperature changes on the surface of the device, the selection of temperature measurement elements, heating elements, and other factors. This research paid more attention to the accuracy and precision of the test. For instance, in 1997, IBM (International Business Machines Corporation)’s Alan Claassen et al. [25] compared the characteristics of diode temperature sensors and RTDs utilized for chip temperature measurement in the process of thermal characterization of electronic packages. As illustrated in Figure 5, the two thermal test chip structures are represented. The flip chip-type TTC with diode sensor measures 7 mm × 7.3 mm and integrates five resistive heaters: one is distributed at the periphery of the chip, and four are located in four quadrants. The operation of the heaters is based on the Joule-heating principle, which involves the conversion of electrical energy into thermal energy through the impedance of the resistive material at a constant current. The temperature-sensitive element utilizes 19 diodes and is fabricated through a PN junction process. The temperature measurement is achieved by the temperature dependence of the forward voltage drop at constant current, as predicted by the Shockley equation. The chip is then packaged onto a ceramic substrate using C4 (controlled collapse chip connection) flip-flop soldering technology, with 403 C4 solder balls arranged in a circular pattern in the central area of the chip, forming a 23 × 23 array that decreases at the corners. The RTD-type wire bond TTC measures 11.537 mm × 11.537 mm and is equipped with four four-quadrant-distributed resistive heaters, the heating mechanism of which also relies on the Joule heating effect. The temperature-sensitive elements comprise four spiral-shaped RTDs, which are formed by thin-film deposition and photolithography into a spiral structure with a size of approximately 0.05 mm and a nominal resistance of 10 Ω at 22 °C. This configuration enables high-precision temperature measurement by utilizing the resistance–temperature linear relationship. The chip’s design incorporates reliability test structures, including internal and external perimeter lines for detecting chip cracks and a comb structure for studying corrosion and electromigration. The two chips are packaged in the form of C4 flip-soldered ceramic substrates and lead-bonded MQFP (metal-quadrilateral flat package). With regard to performance, the diode’s linear fit demonstrates a maximum deviation of 1 °C across the range of 22–135 °C, while the quadratic polynomial fit exhibits a sensitivity of −1.7 mV/°C and a capability to measure down to 0.1 °C. The RTD exhibits superior linearity, with a maximum deviation of only 0.1 °C and a sensitivity of 0.45 mV/°C. These two components are utilized in a variety of applications, including electronics package thermal resistance measurements, reliability assessments, and thermal design verification. In 1995, H. Shaukatullah of IBM undertook an evaluation of the current-switching method for determining the temperature of thermal test chips with diodes [26]. This method is relatively simple to use, as it does not require the diode to be calibrated over a temperature range prior to use, as is the case with the constant-current method. However, this method is more susceptible to instrument and measurement errors. The study’s limitations are evident in the small number of tests conducted, the limited data obtained, and the non-representativeness of the results. While the method demonstrates potential for application, further research employing advanced instrumentation is necessary to enhance and refine the procedure.

In the nascent stages of research on thermal test chips, there existed a paucity of unified standards and a dearth of design direction. The design of the chip was often determined by researchers based on their experimental experience and subsequently refined with the findings of their experiments. Consequently, the experiments employing thermal test chips were unable to ensure the universality and reliability of the data, nor could they facilitate the sharing of experimental data. Subsequent to years of experience and preliminary exploration, in 1997, the JEDEC (Joint Electron Device Engineering Council), a division of the U.S. EIA (Electronic Industries Association), summarized the results of the current development and released the JESD51-4 thermal test chip standard [27]. It stipulates a series of criteria for thermal test chip structure performance parameters and design principles, including the following:-The realization of uniform heat distribution, with the heat source area accounting for more than 85% of the total area of the chip.-The placement of temperature measurement components as close as possible to the heat source without affecting the distribution of heat on the wafer.-The use of several types of temperature measurement units, along with an analysis of the advantages and disadvantages of the heating unit and the layout of the structural design of the chip.-The chip thickness, the choice of processing technology, and the design of the fuse, among other considerations.

The advent of the JESD51-4 standard has significantly advanced the research and application of thermal test chips. Subsequent to this development, thermal test chips that adhere to this criterion have come to the fore, accompanied by experimental findings pertaining to package structure testing, thermal simulation parameter extraction, and heat dissipation performance testing employing thermal test chips. As semiconductor processing technology advances, the semiconductor industry experiences rapid development, leading to increasingly sophisticated thermal test chip designs. The design of a thermal test chip necessitates substantial resources, including manpower, material costs, and an extended production cycle, which can impact project schedules. In the pursuit of high performance, developers are also researching low-cost, easily expandable solutions, among others, with the objective of a standard thermal test chip being applicable to the thermal test needs of a variety of situations.

### 3.2. Standard Array Thermal Test Chips

In the context of implementing multiple thermal test requirements within a single chip, the primary constraint is the chip’s physical dimensions. Early thermal test chips were not designed with scalability in mind during layout, and modifying the chip size almost necessitated a complete redesign. To address this challenge, in 2000, A. Poppe et al. employed the basic unit (single instance) design approach and proposed the design concept of IP (Internet Protocol) ization to design multifunctional test chips with scalable dimensions [28]. The core structure consists of repetitive 500 × 500 μm^2^ base cells that can form a variety of arrays. Each cell contains four polysilicon resistors and four MOS (metal oxide semiconductor) transistor switches, and the switch states are controlled by boundary scan or external signals to realize programmable power consumption modes. The unit power consumption is measured at 80 mW, with a power density of approximately 0.32 W/mm^2^ and a heating area coverage of 87%, thereby meeting the JESD51-4 standard [27]. The temperature-sensitive element employs a CMOS frequency output type sensor, distributed in the center and edge of the unit, supporting serial scanning or parallel measurement. It possesses a temperature resolution of 0.02 °C and a time resolution of 0.5 ms. The chip layout is manufactured using a standard CMOS process, supporting MPW (multi-project wafer) and physical splicing technology. It can be scaled up to a size of 24 × 24 mm^2^. The functionality encompasses thermal transient testing, steady-state temperature distribution analysis, and the absence of external tester operation, rendering it suitable for evaluating the thermal characteristics of a wide range of packages. IP-based design solutions are widely used in current thermal test chips; however, the problem of interconnecting multiple units was not addressed by A. Poppe et al., and no subsequent research has progressed in this area.

In 2008, the TEA (Thermal Engineering Associates) in the United States initiated the design and development of a standard thermal test chip [29]. The chip’s design is predicated on the fundamental unit cell structure, with its core composition comprising metal film resistors and diode temperature sensors, as illustrated in Figure 6. Each base unit integrates two metal film resistors, with a nominal resistance value of 11 Ω. The heating element occupies 86% of the effective area within the unit, in strict compliance with the JESD51-4 standard on the proportion of the heating area requirements. The four-wire Kelvin connection is employed to eliminate contact resistance during measurement of the interference. The metal film resistor exhibits an in-wafer resistance deviation of no more than ±5%, with a deviation within the 4 × 4 array controlled to ±2%. It possesses a low temperature coefficient characteristic, ensuring the stability of power output. The temperature sensing module consists of four diodes, respectively, laid out in the center of the unit, the diagonal corners, and the center of the edge of the position. The module is driven by a 1 mA constant-current source, generating a forward voltage of 0.7 V at 25 °C. Ensuring the measurement’s accuracy necessitates the calibration of the diodes with a K-factor, which has a nominal value of −0.5 °C/mV and a standard deviation of less than 1% post-calibration. Thus, the temperature measurement sensitivity is −2 mV/°C for a current of 1 mA. The fundamental unit size is designed to be 2.54 mm × 2.54 mm, with the capacity for flexible expansion to 3 × 3 to 9 × 11 and other array configurations. The array edge is set up with a width of not less than 150 μm of non-heated area, encompassing the cutting channel, bonding area, and crack suppression structure. The manufacturing process utilizes metal film deposition to form the resistor layer and implements a two-stage wafer dicing process. The initial dicing stage removes the metal interconnect layer, while the subsequent dicing stage completes the grain separation, thereby mitigating the risk of short-circuiting caused by metal residues. Performance tests demonstrate that the maximum power-carrying capacity of a single resistor is up to 11 W. By adjusting the configuration of arrays, it is possible to simulate 65% to 100% coverage of the heating area, with a temperature gradient of approximately 6 °C between the unheated edge area and the core area under full-power conditions. The temperature measurement system utilizes a four-wire connection and K-factor calibration to achieve high-precision data acquisition. The chip’s versatility extends to its applications in thermal resistance analysis, power distribution mapping, and local heat dissipation technology research.

It is evident that the chip design facilitates the modeling of application chips of varying sizes by adjusting the layout of the cells, thereby reducing the cost and time associated with the creation of a thermal test chip of a specific size. In comparison with previous thermal test chips, the thermal test chip designed by TEA generates a relatively uniform thermal distribution within the cell, significantly reducing the production cost. The chip’s simple structure and established process also render its thermal model straightforward to implement. The chip’s thermal model can be integrated with other models for simulation purposes. The chip’s layout design incorporates a symmetrical structure, with the pad area of the internal components’ lead locations being significantly larger. This ensures that over-scribed positions are connected to neighboring cells, facilitating the implementation of resistors in series and diodes in row-selected connection between different cells on the wafer. Prior to the scribing process, it is imperative to undertake a de-metallization procedure on the wafer. Furthermore, by manipulating the position of the scribe line, the chip can be efficiently shaped into an array size, thereby achieving substantial cost savings.

In summary, the TEA study provides a framework for the design and packaging of thermally tested chips. However, it does not include comparative work on the temperature distribution of actual chips, nor does it address repetitive and reliability testing of the basic unit and various different arrays.

Whilst general-purpose thermal test chips are capable of meeting the majority of testing requirements, there are still instances where specialized thermal test chips are required for thermal testing. In 2006, X. Jordà et al. from the IMB-CNM (CSIC) (Institut de Microelectrònica de Barcelona, Centre Nacional de Microelectrònica) (Consejo Superior de Investigaciones Científicas) utilized a specialized thermal test chip to simulate the thermal state of power devices [30,31]. As illustrated in Figure 7, the proposed TTC employs a 6 mm × 6 mm × 0.525 mm silicon-based architecture, integrating 130 parallel polysilicon heating resistors on the top surface, with a single strip width of 20 µm and a pitch of 17.2 µm. This configuration covers 98% of the effective area, thereby ensuring a uniform heat-flow distribution and a maximum power density of 170 W/cm^2^. A platinum resistance temperature sensor measuring 700 µm × 700 µm is integrated in the central area, exhibiting a sensitivity of 0.95 °C/Ω and a temperature accuracy of ±0.5 °C over the range of 20 °C to 200 °C via four-wire measurements. The fabrication of the chip entails a series of processes, including polysilicon deposition and etching, platinum sensor photolithography, and Ti/Ni/Au metallization on the backside, ensuring compatibility with standard power device packaging processes. The TTC is designed for applications such as power substrate thermal resistance evaluation, dynamic thermal simulation verification, and calibration of high-precision temperature measurement systems. FLOTHERM simulation has been employed to verify the heat flow uniformity of the TTC, and the isothermal surface below 38 µm from the top is found to be parallel to the bottom surface of the chip. This observation aligns with the simplification requirements of vertical heat conduction models.

In 2013, Toshihiro Matsuda et al. [32] designed a thermal test chip structure for testing various packages in order to analyze transient thermal phenomena on LSI (large-scale integration) chips. The chip is fabricated using a standard 0.18 µm CMOS process, and the overall architecture consists of 24 sensor modules arranged in 4 rows and 6 columns, each measuring 1.23 mm × 1.23 mm. As shown in Figure 8, each module integrates a heater and 32 temperature-sensitive elements arranged radially around the heater. Among them, two types of p-type diffusion resistors and p-type polysilicon resistors are used for the heating element, and the dimensions of the two resistors are 10 μm × 40 μm and 10 μm × 13 μm, respectively, and their resistances are designed to be in the order of 500 Ω. The temperature-sensitive elements are designed as follows. Controllable heating is performed by applying a heating power of 50 mW to 110 mW to generate the Joule effect. The temperature-sensitive element is constructed based on p-n junction diode devices, and the temperature detection is realized by the forward voltage temperature effect at a constant bias current of 10 µA, and the Kelvin connection architecture and X/Y decoder addressing technique are used to realize high-precision electrical measurements. Experimental data show that the temperature field distribution is inversely proportional to the distance L from the heating source and that the thin top-layer chip with a thickness of 50–100 µm exhibits a significant temperature gradient enhancement due to the thermal resistance of the Henkel QM1536 bonding material. The test chip is capable of transient thermal analysis with 10 µs time resolution, and the consistency of the temperature distribution pattern with the experimental measurements is verified using the Ansys Icepak 14.0 thermal simulation platform.

In 2006, Teck Joo Goh et al. [33] designed a thermal test chip with a specialized heater structure, known as the “fireball heater”, for the study of hot spots on the chip. The structural design of the TTC centers on a serpentine metal resistance heater and a fireball heater. The serpentine heaters are connected in parallel or series with multiple legs for resistance adjustment and cover ≥85% of the chip area to meet the JEDEC guideline for uniform heat generation. The fireball heater, with a size of approximately 1/20th of the chip area and a power density of >1 kW/cm^2^, is used to evaluate the thermal diffusion performance of TIMs (thermal interface materials) and heat dissipation schemes. The temperature sensor uses a metal RTD that eliminates wire resistance errors through a four-wire measurement method, with a size of <100 × 100 μm^2^, a power density of <0.1 W/cm^2^, and a temperature measurement accuracy of up to 0.06 °C. In the chip layout, the temperature sensor RTDs are distributed at the edges, diagonally, and near hot spots. The manufacturing process is based on a silicon process with an integrated multilayer metal structure. In terms of performance, the conventional test power density is >100 W/cm^2^, and the heat flow density is verified for uniformity by finite element analysis, which shows that the MC (metal coverage) ≥ 50%, ensuring the uniformity of temperature distribution. The chip is applied to thermal resistance measurement, assembly optimization, product aging simulation, and TIM performance evaluation, and the fireball heater is especially suitable for developing high-precision heat dissipation solutions. In 2012, B. L., Lau et al. at the Institute of Microelectronics, Singapore, designed a silicon-based thermal test chip with a power density of 11.9 kW/cm^2^ [34], dedicated to the development of cooling solutions for hot spots in GaN-on-Si high-power devices and thermal characterization. The chip consists of a highly doped n-type resistor with a p-n junction-based temperature sensor, with an overall size of 7 mm × 7 mm × 0.2 mm. The heater is designed in multiple sizes, with individual heater sizes ranging from 40 μm × 350 μm to 450 μm × 5280 μm, to generate a localized heat flux density of up to 11.9 kW/cm^2^ through the Joule effect while maintaining an average thermal load at the chip level of 100 W/cm^2^. Temperature sensors are distributed 5–10 μm from the heating element, and temperature measurement is realized by the negative temperature coefficient characteristic of the forward voltage of the p-n junction, with a sensitivity of −30 mV/°C and a temperature range of 25–200 °C. The chip fabrication process includes ion implantation, high-temperature annealing, PECVD (plasma-enhanced chemical vapor deposition) dielectric deposition, and metallization. The electrical performance tests of the resistors and sensors show that the standard deviation of their resistance values is 0.1–3 Ω, verifying the uniformity and repeatability of the process. The functions of this thermal test chip include verifying the thermal performance of the microchannel cooler and supporting the research of thermal management technology in the field of high-power electronic packaging. The thermal test chip layout and heater design are shown in Figure 9.

Following years of development, thermal test chips have become ubiquitous in the field of chip thermal testing. Initially employed as heat sources and sensors for material thermal property testing, these chips have evolved into a pivotal medium for thermal modeling and verification of thermal simulations. Chip thermal design continues to be predominantly characterized by modeling and simulation, with the majority of research in the field of thermal test chips being integrated with thermal simulation. However, the prevailing chip model remains relatively elementary, exhibiting a lack of adaptation to the intricate, parameterized designs characteristic of modern chip architecture [35]. To address the intricate thermal environment of the chip and to study its fine-grained model, the requirements for the performance of thermal test chips have been elevated. This necessitates enhanced thermal power control accuracy, more comprehensive temperature data, and higher spatial resolution. In 2019, the JEDEC released the JESD51-4A standard [36], which is based on the previous standard and considers power mapping as the primary thermal test chip objective.

### 3.3. Programmable Control Thermal Test Chips

Presently, the focus of research in the field of thermal test chips is primarily oriented towards the analysis of complex thermal environments. These environments demand a heightened spatial resolution in the distribution of heating and temperature measurement elements. This is to ensure that the production of fundamental elements in a given unit area can satisfy the requirements of the thermal model. Additionally, there is a need for enhanced control accuracy in thermal power to facilitate the completion of intricate power mapping. Furthermore, there is a demand for increased reading speed and transient temperature sensitivity. The advent of integrated circuit processing has led to a marked proliferation of temperature measurement and heating elements on the thermal test chip, with the number rapidly escalating from dozens to nearly a thousand. This augmentation in the quantity of fundamental components has concomitantly given rise to increasingly intricate circuit connections, thus prompting research to concentrate on programmable control.

A programmable-driven thermal test chip was designed at the IMEC (Interuniversity Microelectronics Centre) in 2015 [37,38]. As shown in Figure 10, the chip size is 8 × 8 mm^2^ and consists of 4 × 4 basic cells, each of which contains 8 × 8 square cells with a square cell size of 240 × 240 μm^2^. Each cell contains a centrally located thermal diode as a temperature sensor, enabling 32 × 32 thermal pixel imaging. The single diode temperature sensor measures approximately 4.8 μm × 2.6 μm and is fabricated using FEOL (front-end-of-line) semiconductor processing technology. Each base cell contains two 200 μm × 100 μm metal meander heaters as heating units, fabricated using BEOL (back-end-of-line) technology, and individually controlled by each of the two transmission gates. The chip is fabricated in a CMOS process, combined with thermal compression bonding and underfill technology for 3D stacking, and packaged in an fcFBGA (flip chip fine-pitch ball grid array). The entire thermal test chip has 832 individually addressable heater cells and 1024 temperature sensor cells, with 75% of the active area. The chip’s heat flow density can attain 10 W/cm^2^, the diode temperature sensor’s calibration sensitivity can reach 95%, the temperature measurement range is from 10 °C to 75 °C, and the temperature measurement sensitivity is −1.55 ± 0.02 mV/°C for a current of 5 µA. It is evident that the thermal test chip manufactured by IMEC exhibits both high spatial resolution and high power density. The chip’s programmable power distribution capability enables its adaptability to a wide range of complex thermal test experiments.

In 2017, Xilinx published a report on the structure of a silicon chip intended for use in a thermal evaluation system [39,40]. As shown in Figure 11, the thermal test chip adopts a two-dimensional array structure, with each cell consisting of 20 × 20 small cells, each of which integrates FEOL resistors and ring oscillators as programmable heating elements, supporting row-by-row or bank-by-bank independent control with a maximum power density of 3 W/mm^2^. The resistor generates heat through the Joule effect, and the ring oscillator generates heat through capacitor charging and discharging and switching frequency. In addition, each heating unit has temperature sensors, which include diodes, resistors, and ring oscillators, and its temperature signal is converted to a digital signal by a digitizing circuit [41], the principle of which is shown in Figure 12. The calibrated temperature measurement accuracy is ±2 °C, and the temperature measurement range is 30–125 °C. The chip is fabricated using a 0.18 μm CMOS process, and the FEOL resistor and ring oscillator are integrated in the front layer of the chip. The temperature sensor uses the same process, with the diode, resistor, and ring oscillator co-integrated with the heater. The package is a flip-chip package to support low-cost digital interface testing. In the meantime, the chip can also be used for thermal optimization of chip layout, calibration of thermal simulation models, evaluation of package thermal solutions, and dynamic thermal analysis of customer use cases.

The chip’s high degree of digitization enables the generation of complex thermal distribution arrays, facilitating a straightforward and controllable test operation. The mapping scheme can be adjusted on-site, and the circuit possesses a robust thermal evaluation capability. However, it should be noted that thermal testing typically necessitates the use of extreme thermal environments, and digital circuits often exhibit inadequate temperature resistance, necessitating the implementation of effective heat dissipation schemes.

Wen Yueh et al. [42] in 2016 proposed an on-chip digitally programmable thermally coupled simulation framework called FPTE (field programmable thermal emulator) for online characterization of power maps and time-varying thermal fields. It also facilitates the analysis of changes in transistor electrical characteristics. The FPTE employs a digital heater and sensor array, based on a CMOS process, to achieve highly accurate thermal field simulation and on-line characterization. Five symmetrically distributed FPTE cores, each with an area of 0.0375 mm^2^ and a fill factor of 50%, are integrated on a chip size of 2 mm × 1 mm. The heating element consists of n-well resistors and NMOS (N-channel metal oxide semiconductor) transistors, including 4 × 4 arrays of sub-resistor blocks, each with dimensions of 60 μm × 50 μm and a total equivalent resistance of 250 Ω. The control of the heating element is facilitated by a 4-bit register comprising 16 stages of power outputs, with a maximum power of 544.5 mW and a power density of up to 1452 W/cm^2^. The layout of the coaxial structure is designed to optimize thermal homogeneity, while the concentric structure is employed to enhance thermal uniformity. The temperature sensors employed in this system include a BJT-type (bipolar junction transistor-type) analog sensor and an RO-type (ring oscillator-type) digital sensor. The BJT-type analog sensor output is quantized by an external ADC (analog-to-digital converter) with a resolution of 0.4 °C, and the RO-type digital sensor consists of a 9-stage ring oscillator and a 32-bit counter with a sampling frequency of 500 MHz and a resolution of 0.303 °C. The fabrication of the FPTEs is performed on a 130 nm CMOS process that is compatible with standard processes. Currently, the FPTE has been successfully applied to multi-core thermal coupling analysis, package thermal resistance evaluation (fluid cooling thermal resistance 26.9 K/W), and dynamic thermal field prediction (correlation coefficient 0.9846). It is important to note that the FPTE is not a thermal test chip in the traditional sense. However, the FPTE’s functionality is analogous to that of a TTC in that it is capable of generating controlled, time-varying, arbitrary power consumption patterns. Experimental results demonstrate its capacity to generate dynamic sensor outputs that can be converted into temperature data, thereby characterizing the thermal coupling between cores.

In 2021, Romina Sattari et al. [43,44,45] at Delft University of Technology conducted research on a thermal test chip containing two metal layers. The first layer was a 100 nm thick titanium film used to create the microheater and RTD, and the second layer was a 2 μm thick aluminum film used to add a single bump measurement cell and daisy-chain connections. As illustrated in Figure 13, the chip’s core unit measures 4 × 4 mm^2^ and features a modular layout that facilitates arbitrary array expansion. Each cell contains six independently controlled micro-heaters and three RTDs. The micro-heaters achieve a uniform thermal distribution with a power density of 100 W/cm^2^ through the Joule effect, and the RTDs utilize a four-point Kelvin connection. The linear and spiral RTDs provide a temperature sensitivity of 12 Ω/K and 9 Ω/K, respectively. The fabrication of the chip entails a bimetallic layer process, initiated by the sputter deposition of a 100 nm titanium thin film layer on a 4-inch p-type wafer with a thickness of 525 μm. Subsequent steps include photolithography and RIE (reactive ion etching) to form the micro-heater and the RTDs. Interconnections and bump structures are realized by sputtering a 2 μm aluminum layer, and finally subsequent processes such as deposition of the passivation layer, contact openings, and pad metallization are completed in sequence. Experimental findings demonstrate that the TTC exhibits stable temperature measurement performance within the range of −55 °C to 150 °C, with a heating area coverage as high as 82.5%, and the uniformity of thermal distribution is verified by infrared thermography. The chip finds application in power package reliability assessment, thermal cycle testing, and in situ monitoring of automotive electronics, providing a flexible platform for thermal management research of high-power-density electronic devices.

Concurrently, the development of commercialized thermal test chips is undergoing rapid advancement, as evidenced by the emergence of TEA’s TTC-1001 and TTC-1002 series chips and NANOTEST’s NT16-3k [46] and NT20-3k [47] series chips, among others. These chips prioritize compatibility with the process, cost-effectiveness, and the precision of the thermal model. In comparison with the development of foreign thermal test chips, the domestic focus has been more on the commercialization of thermal test chips for thermal testing, with less emphasis on the research and design of thermal test chips. In 2021, SMIC (Semiconductor Manufacturing International Corporation) [48] developed a thermal test chip for FCBGA packages. The chip utilizes a metal strip resistor as the heating element, with 15 heaters arranged in a 5 × 3 array with a basic heating element size of 4.1 mm × 5.0 mm. Each heater is connected via a four-wire connection to achieve precise Joule thermal control. The temperature-sensitive element consists of seven RTDs with a basic element size of 0.2 mm × 0.2 mm, distributed at the center, edges, and corners of the chip. A four-wire system is also employed to improve measurement accuracy. The chip’s dimensions are 25 mm × 16 mm, with a top metal layer that integrates the heaters and sensors and a peripheral RDL (redistribution layer) for daisy-chain reliability testing. Subsequent analysis indicates that the power density of the entire chip can reach 0.8 W/mm^2^, with the local area reaching 5 W/mm^2^. However, it is important to note that the chip has been designed and thermally modeled solely for the purpose of simulating the thermal test chip, and it has not been physically fabricated.

To summarize, the prevailing thermal test chip design employs a cell array structure. Due to the uncomplicated resistance process, these elements have become prevalent heating mechanisms. Temperature measurement devices encompass thermistors, RTDs, diodes, and digital circuits utilizing ring oscillators, among others. RTDs and diodes exhibit simplicity and reliability in their operation; however, they necessitate supplementary drive circuitry, resulting in a more intricate interconnection pattern. Conversely, digital circuits boast advantageous features such as immunity to interference, rapid acquisition, and other salient characteristics. However, their high temperature resistance is inadequate, their process is more intricate, and they are not conducive to adjustment or expansion. The prevailing focus in the current stage of research on thermal test chips is on the realization of complex power mapping and temperature and stress tests. The following table (Table 1) organizes some representative thermal test chip parameters.

## 4. Advances in Thermal Test Chip Application Research

The thermal test chip is a tool that enables researchers to accurately simulate the heat generation under actual working conditions. This capability facilitates a comprehensive understanding of the internal heat and stress patterns of the device. Additionally, it allows for the evaluation of the heat and stress distribution in different locations and the adjustment and optimization of the package structure accordingly. Currently, both domestic and foreign researchers are employing the thermal test chip in a variety of research areas, including chip temperature distribution, interface thermal resistance, and chip heat dissipation.

### 4.1. Characterization of Chip Temperature Distribution

In 2013, the TEA Association undertook a study on the application of a designed thermal test chip in stacked chip packaging [49]. There are two broad categories of packages for single and multi-chip: lead-bonded packages and flip chip packages. For the WB (wire bond) version, the base chip is adhered to the package substrate by a conductive adhesive, and the top chip is held in place using a non-conductive material. Wire bond pads on the chip facilitate connection, while the FC (flip chip) version utilizes lead-free solder material with solder bumps measuring 0.169 mm in diameter and 0.1 mm in height. The top chip is fixed by non-conductive material, and the detailed dimensional configuration is depicted in the following table (Table 2) and Figure 14. In both package versions, polycarbonate covers are installed after bonding to protect the chip and lead bonding assembly. Subsequently, thermal models were constructed for temperature distribution simulations for single-chip packages and multi-chip packages, respectively, and were compared with the measured values of the thermal test chip. The significance of this study lies in the use of a thermal test chip and modeling software to reveal information about the interior of the stacked chips and to present a matrix method to evaluate the temperature distribution in single and multiple chip assemblies.

In 2012, MICROSANJ published a comparison of results on IR imaging and thermal reflection imaging [50]. A parallel comparison of TR (thermal reflection) imaging and IR (infrared reflection) imaging was performed using a specially designed thermal test chip with embedded diode temperature sensors (CMOS thermal test chip TTC-1002). The TTC-1002 chip contains two heaters made of titanium in each cell and five embedded diode temperature sensors distributed at the center and its different corners. The active area is 86% of the area. The spatial, thermal, and temporal resolutions of the two imaging methods were investigated, and the results were verified with the integrated diode temperature sensors. The findings revealed that thermal reflectance imaging exhibited superior spatial resolution, temporal resolution, and temperature accuracy on metal heaters. In contrast, infrared imaging demonstrated reduced accuracy on metals that were not coated to enhance emissivity. The thermal reflectance imaging measurements were found to be within 1.7% of the diode readings, while infrared imaging was within 6%.

In 2015, IMEC characterized the thermal performance of 3D package structures using a designed programmable driven thermal test chip [37]. The chip was subjected to both uniform and localized hot-spot power distributions, and its internal temperature distribution was tested under various cooling conditions. This investigation was undertaken to examine the self-heating effect, electro-thermal coupling, and lateral diffusion of on-chip temperature within the 3D chip. The characterization results were then verified using a thermal finite element model, as illustrated in Figure 15. The validation results demonstrate a substantial overall agreement between the finite element simulation outcomes and the measured characterization results, which is most significant at the local temperature peaks. As demonstrated in Figure 16, considering the molded plastic package in the low-power interface as a case study, the temperature distributions of the top and bottom chips after calibrating the boundary conditions exhibit a high degree of agreement with the simulation results, with the maximum temperatures recorded at approximately 38 °C. The temperature distribution of the top and bottom chips after calibrating the boundary conditions is in good agreement with the simulation results. However, it should be noted that due to the material properties inherent in the BEOL, there is a 5–10% prediction deviation in the simulation results. Furthermore, the implementation of the stack-level compact model results in an overestimation of the chip edge temperature, suggesting that the lateral heat diffusion within the package is not sufficiently captured by the equivalent convection boundary conditions employed in the compact model.

### 4.2. Integrated Process Testing

In 2012, Tsinghua University developed a thermal test structure employing TEA’s thermal test chip, designated as TTC-1002, to assess the impact of temperature on the performance of TSV (Through Silicon Via) [51]. Each resistor in the thermal test chip exhibited a nominal resistance of 7.6 Ω ± 10%. The nominal forward voltage of each diode was 0.71 V at a forward current of 1 mA, and the reverse voltage was 7 V at a reverse current of 10 μA. As demonstrated in Figure 17, the thermal test chip incorporated diodes for the calibration of the junction temperature. The assembly utilized four thermochips, with a total size of 5.8 mm × 5.8 mm × 0.64 mm, in conjunction with two parallel resistors for each test. The silicon inserter measures 14 mm × 14 mm × 0.1 mm, with a TSV diameter ranging from 15 to 25 μm and a depth-to-width ratio of 5:1. The TSV is filled using a damascene copper plating process, and the silicon inserter fabrication process utilizes a double-sided copper RDL. The linear characteristic between forward voltage and junction temperature is calibrated by using a thermostat bath environment, which allows for a temperature accuracy of ±0.1 °C. The TSV is calibrated by using a thermostat bath.

In 2006, X. Jordà et al. from the IMB-CNM (CSIC) proposed a methodology for in situ measurement of the thermal conductivity of the dielectric layer based on a thermal test chip designed to simulate and derive the thermal resistance of two typical power assembly structures based on two substrates—IMS-1, IMS-2 (insulated metal substrate-1, insulated metal substrate-2)—and a thermal test chip [52]. The IMS-1 and IMS-2 structures both contain a 100 μm copper layer, a dielectric layer, and a 2 mm aluminum substrate, with the difference that the thickness of the IMS-1 dielectric layer is 100 μm, while the thickness of the IMS-2 dielectric layer is 75 μm. The power dissipated by the chip under simulated and measured conditions is 30 W, and the ambient temperature is room temperature. The corresponding experimental results validate the simulation results and thus the proposed thermal conductivity extraction method, as shown in Figure 18. The experimental results show that the simulated and experimental thermal resistances of IMS-1 are 2.55 K/W and 2.48 K/W, respectively, and the simulated and experimental thermal resistances of IMS-2 are 2.07 K/W and 2.03 K/W, respectively. The thermal conductivities of the dielectric layers of IMS-1 and IMS-2 are 1.25 W/m·K and 1.20 W/m·K, respectively.

In 2022, Romina Sattari et al. at Delft University of Technology used a self-developed bimetallic layer thermal test chip to characterize and compare the thermal resistance of silver and copper sintered molded joints using the TDIM (transient dual interface method) [45], as shown in Figure 19. The developed thermal test chip has a heating area fraction of 82.5%, a temperature sensitivity of 12 Ω/K, and a maximum power of 360 W. Meanwhile, infrared thermography verifies the uniformity of the temperature distribution (less than 1 °C temperature change under 68 W power excitation). The junction thermal resistance of the sintered structure was extracted by using the Ag and Cu sintering method to combine the thermal test chip on a Cu substrate. The experimental results show that the lowest thermal resistance of the sintered structure with Ag paste is 0.144 K/W. The thermal resistance of the Cu sintered structure is 0.158 K/W. The thermal resistance of the Cu sintered structure is 0.158 K/W.

### 4.3. Chip Heat Dissipation Research

Experimental characterization and model validation of liquid jet impingement cooling were performed at IMEC 2019 using an in-house developed thermal test chip [38]. The article modeling and measurement studies were applied to two implementations of jet impingement cooling: (1) a single-jet cooler with a diameter of 2 mm and (2) a multi-jet cooler with 4 × 4 rows of 500 µm inlet nozzles and distributed outlet nozzles. For both cooling configurations, temperature measurements and CFD (Computational Fluid Dynamics) modeling results were taken and compared for uniform and hot spot power dissipation patterns as shown in Figure 20 and Figure 21. The experimental results show that the multi-jet impingement cooler has lower thermal resistance and better temperature uniformity than the single-jet cooler at the same flow rate. For the 4 × 4 multi-jet array cooler, when the flow rate is 600 mL/min and the required pump power is 0.4 W, a very low thermal resistance of 0.25 K/W can be achieved.

In 2024, a team from the BUAA (Beijing University of Aeronautics and Astronautics) experimentally analyzed the start-up behavior of a manifold microchannel radiator in a mechanically pumped flow circuit using the dynamic response characteristics of the thermal test chip temperature [53]. An analog thermal test chip embedded with a temperature sensor was used as the heat source to provide fast and accurate temperature measurements in the heated zone. Using ammonia as the coolant, a heat flow of 634 W/cm^2^ was effectively dissipated over a chip area of 6 × 6 mm^2^, as shown in Figure 22. Based on the dynamic response characteristics of the chip temperature, the start-up process was categorized into three types: gradual, transition, and overshoot. During the startup process, the junction temperature increased to 86 °C and finally stabilized at 75.1 °C, while it was observed that the transient maximum chip temperature during the startup process could exceed the final steady-state temperature by nearly 40 °C, and the higher flow rate and subcooling inlet could mitigate the temperature overshoot under high heat flux conditions. The near-saturated inlet facilitates a smooth transition between flow regimes, which is advantageous at moderate heat fluxes.

To summarize, domestic and foreign researchers have conducted studies on chip temperature distribution, interface thermal resistance, and chip heat dissipation using thermal test chips. The representative thermal test chips utilized in these studies include the TTC-1002, which was designed by TEA, and the programmable driver thermal test chip, which was designed by IMEC. The following table (Table 3) delineates the parameters and specific applications of the aforementioned thermal test chips in the context of the aforementioned application studies.

## 5. Summary and Prospects

At present, the thermal test chip gradually formed a standard general-purpose chip according to the JEDEC51-4A design standard, TEA, Nanotest company as a representative, has introduced a general-purpose thermal test chip and applied it to chip package design and process optimization, heat dissipation, and other research areas. However, at present, only a single temperature field or stress field test can be realized, and a single test chip that can test temperature, stress, electromigration, and other multi-physical field parameters has not yet been realized, thus limiting the scope of its application. The development of Chiplet, advanced packaging technology, promotes the development of microsystem technology. Microsystem structure is becoming more and more complex, and the interface within the system multi-physical field coupling problem is becoming more and more prominent. In addition to measuring temperature parameters, we also need to measure the stress, electromobility, electromagnetic field, and other parameters. The thermal test chip will play an increasingly important role in the design of electronic devices and microsystems. At the same time, it also puts forward higher requirements on the function of the thermal test chip; the thermal test chip must be upgraded from the traditional thermal test to a multi-physical field test. The thermal test chip must be integrated with heating, temperature, stress, electric field, magnetic field, electromobility, and other test components to form a generalized multi-physical field test chip to meet the needs of the development of advanced packaging technology.

## Figures and Tables

**Figure 1 micromachines-16-00669-f001:**
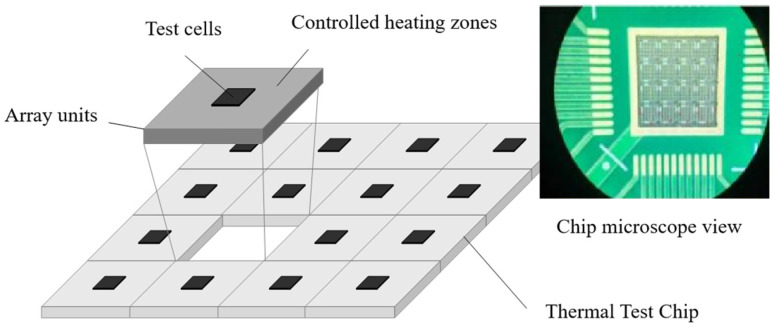
Typical structure of a thermal test chip.

**Figure 2 micromachines-16-00669-f002:**
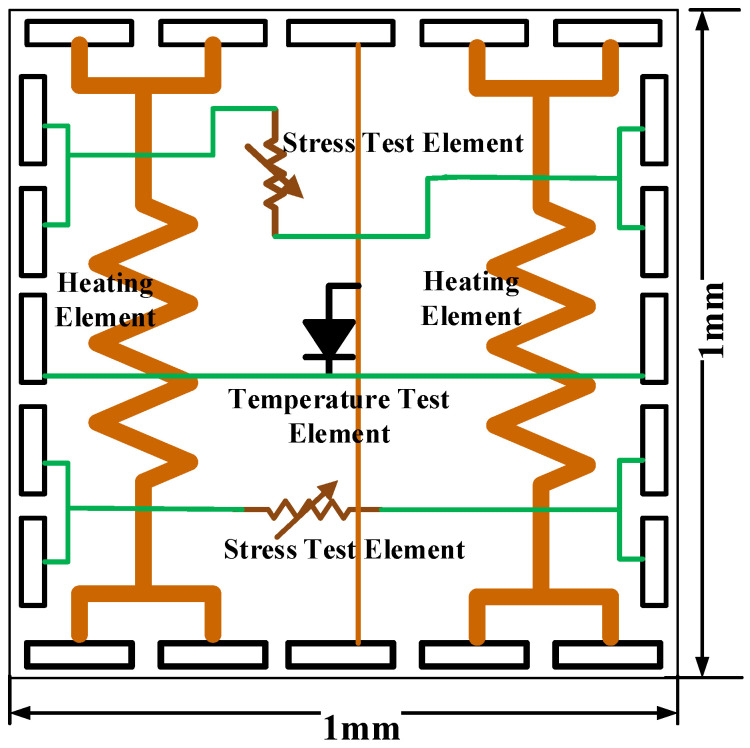
Circuit diagram of thermal test chip unit.

**Figure 3 micromachines-16-00669-f003:**
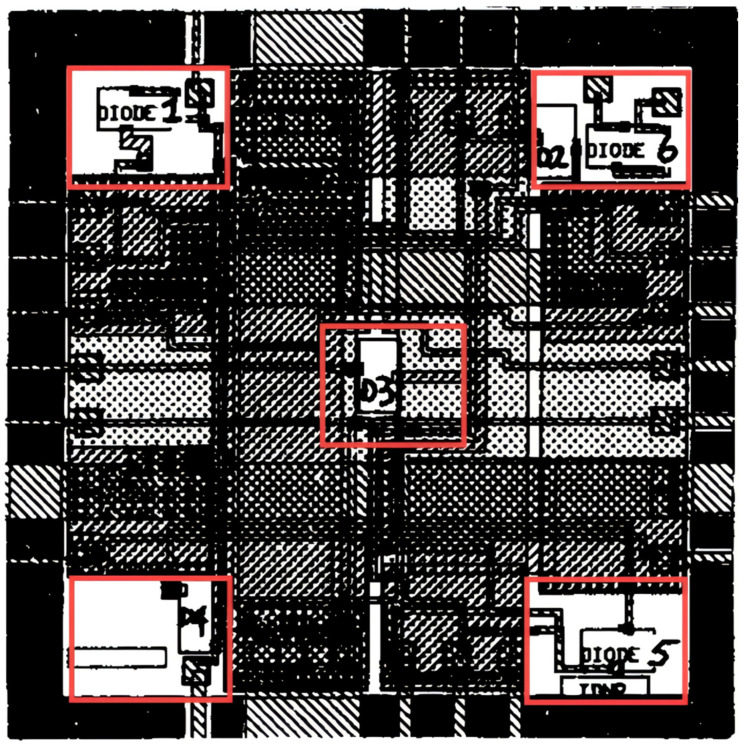
CMOS1 thermal test chip unit (size 2.5 mm^2^, boxed section for temperature-sensitive diodes). Adapted with permission from [20]. Copyright 1989 IEEE.

**Figure 4 micromachines-16-00669-f004:**
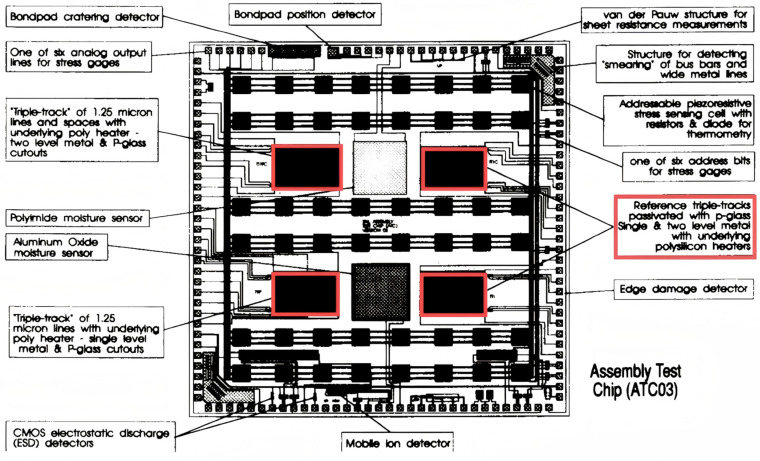
ATC03 thermal test chip layout (boxed section shows four symmetrically distributed polysilicon heating elements). Adapted with permission from [22]. Copyright 1993 IEEE.

**Figure 5 micromachines-16-00669-f005:**
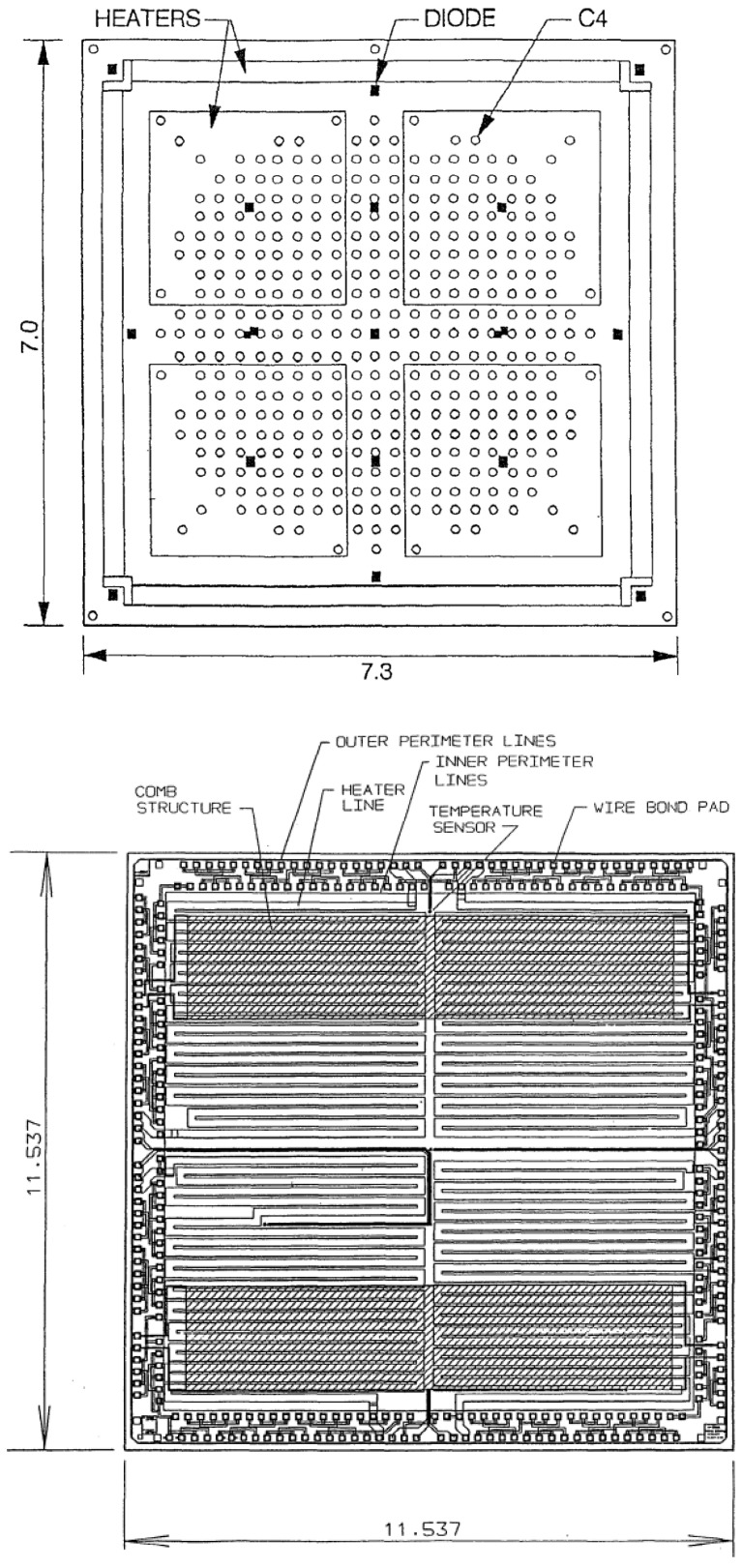
(**Top**) Schematic of TTC using diode temperature sensor; (**bottom**) schematic of RTD and TTC. Reproduced with permission from [25]. Copyright 1997 IEEE.

**Figure 6 micromachines-16-00669-f006:**
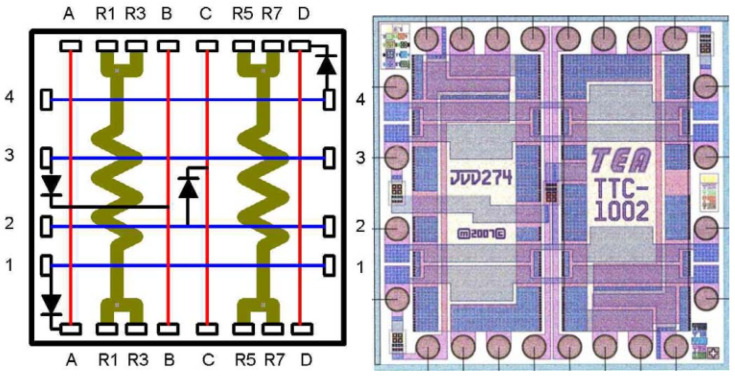
Thermal test chip TTC-1002: (**left**) unit electrical distribution; (**right**) chip resistor and temperature diode layout. Reproduced with permission from [29]. Copyright 2008 IEEE.

**Figure 7 micromachines-16-00669-f007:**
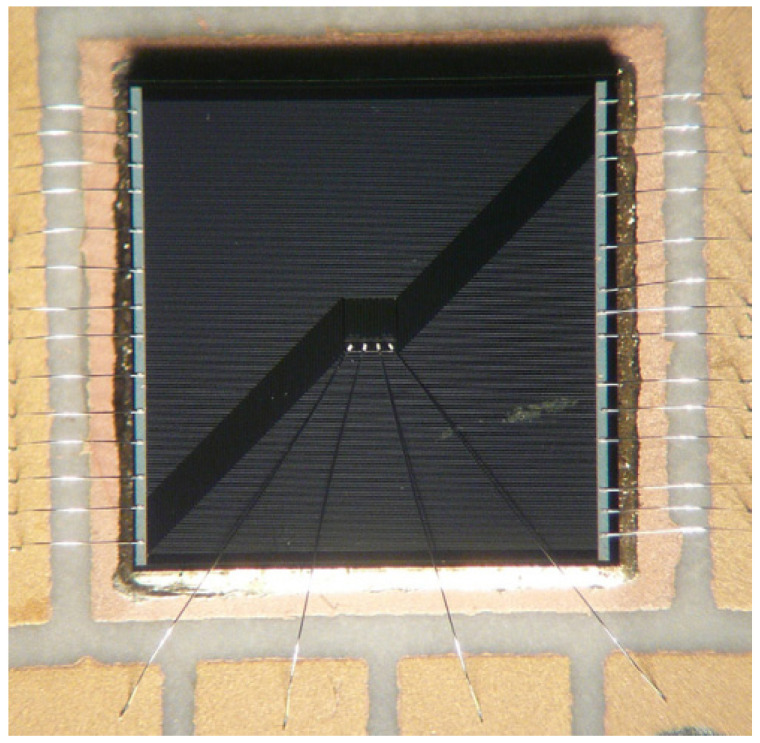
Top view of a TTC soldered on a ceramic substrate. Reproduced with permission from [30]. Copyright 2007 IEEE.

**Figure 8 micromachines-16-00669-f008:**
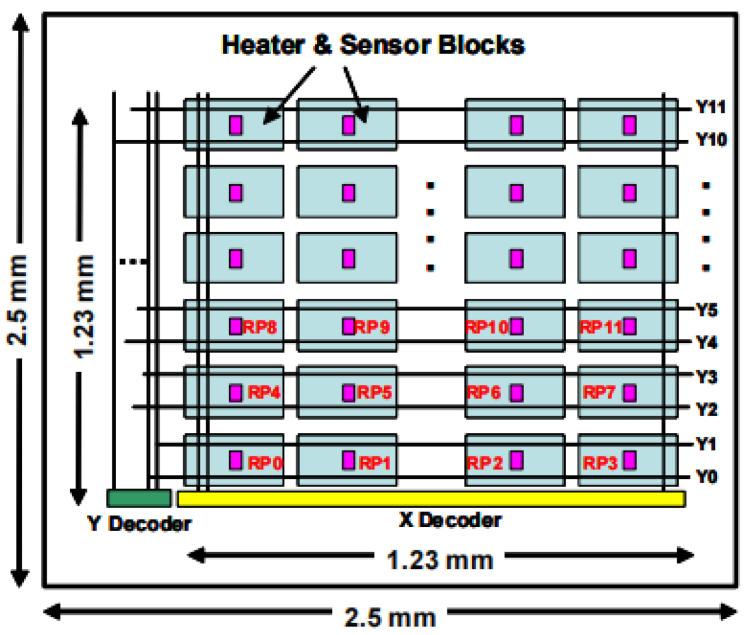
Thermal test chip layout. Reproduced with permission from [32]. Copyright 2014 IEEE.

**Figure 9 micromachines-16-00669-f009:**
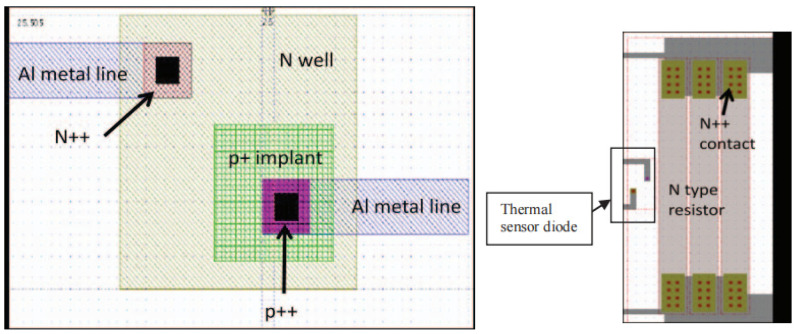
Thermal test chip layout and heater design. Reproduced with permission from [34]. Copyright 2012 IEEE.

**Figure 10 micromachines-16-00669-f010:**
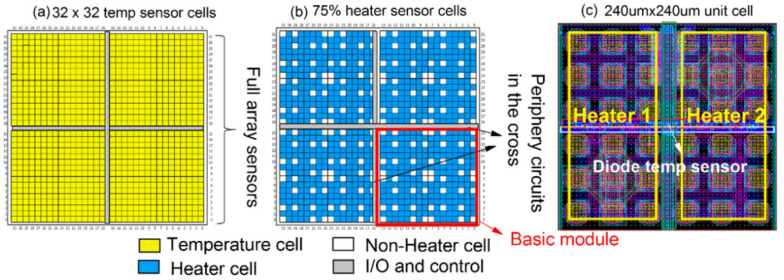
Plan view of an 8 × 8 mm^2^ thermal test chip: (**a**) 32 × 32 temperature sensor array configuration; (**b**) configuration of 832 programmable heater cells; (**c**) detail of a metal curvature heater in a square cell (240 × 240 μm^2^). Reproduced with permission from [38]. Copyright 2019 Elsevier Ltd.

**Figure 11 micromachines-16-00669-f011:**
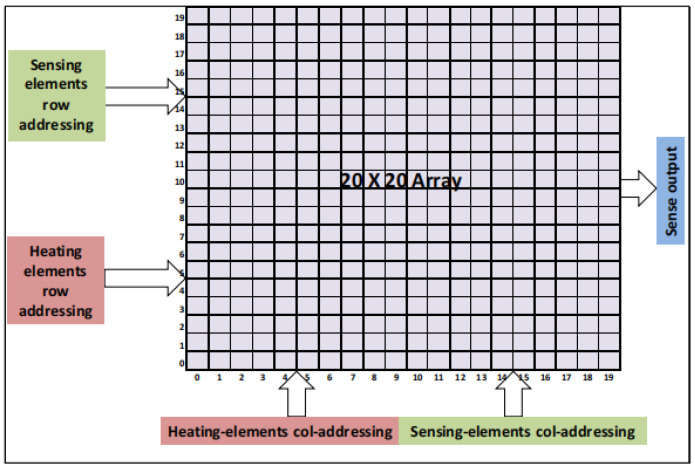
Schematic diagram of Xilinx thermal test chip. Reproduced with permission from [39]. Copyright 2018 IEEE.

**Figure 12 micromachines-16-00669-f012:**
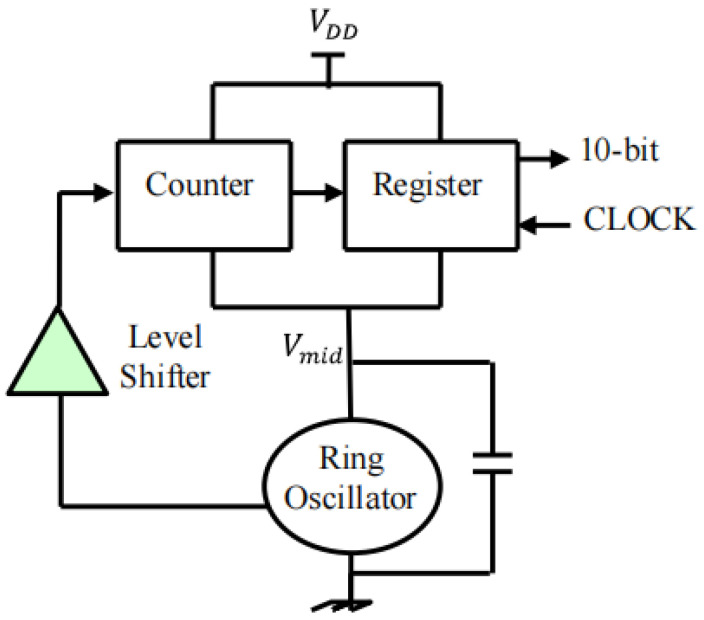
Ring oscillator temperature measurement schematic diagram. Reproduced with permission from [41]. Copyright 2012 [Suman, S.; Singh, B.P.]; distributed under CC BY 4.0 (https://creativecommons.org/licenses/by/4.0/) (accessed on 15 May 2025).

**Figure 13 micromachines-16-00669-f013:**
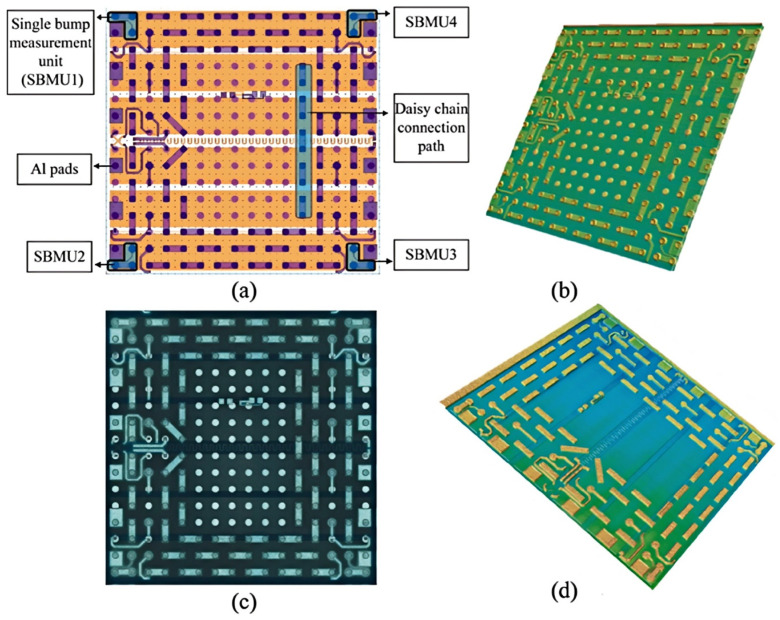
Layout of the bimetallic layer thermal test chip: (**a**) Second metallization layer including SBMU (single bump measurement units) and daisy chain connections; (**b**) 2D (two-dimensional) optical microscopy of a unit cell; (**c**) 3D optical imaging of the complete process fab-out; (**d**) 3D optical imaging after the second metallization layer. Reproduced with permission from [45]. Copyright 2022 Elsevier Ltd.

**Figure 14 micromachines-16-00669-f014:**
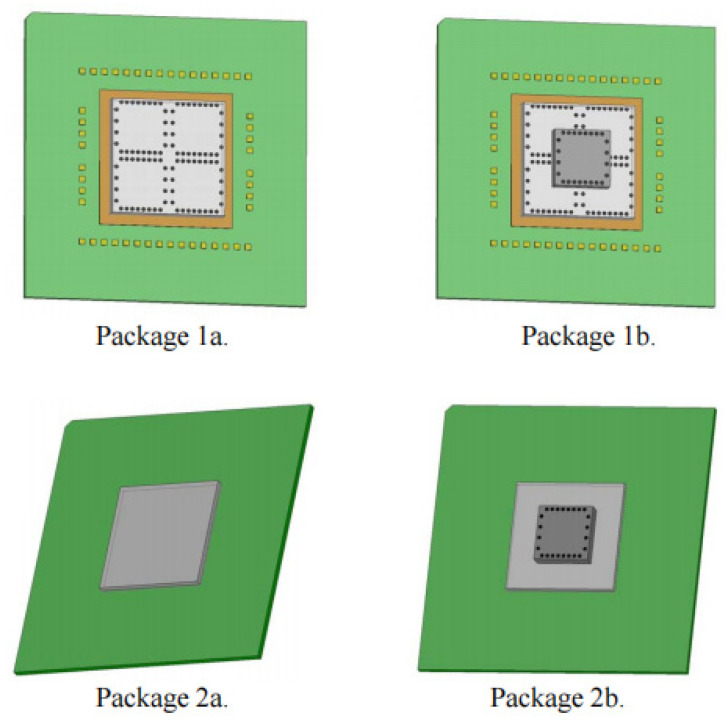
Different encapsulation methods. Reproduced with permission from [49]. Copyright 2013 IEEE.

**Figure 15 micromachines-16-00669-f015:**
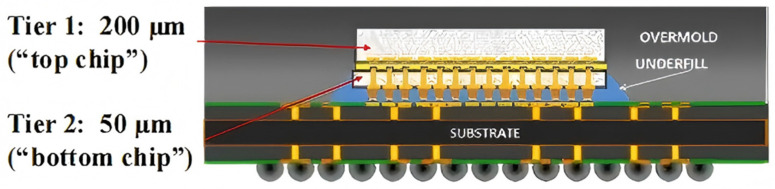
Section of 3D stacked package structure. Reproduced with permission from [37]. Copyright 2015 IEEE.

**Figure 16 micromachines-16-00669-f016:**
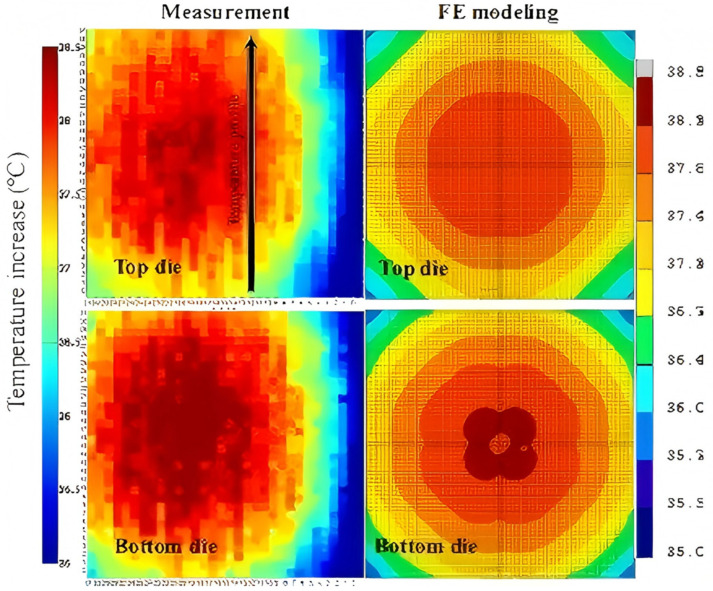
Comparison of measured characterization results (**left**) and finite element simulation results (**right**). Reproduced with permission from [37]. Copyright 2015 IEEE.

**Figure 17 micromachines-16-00669-f017:**
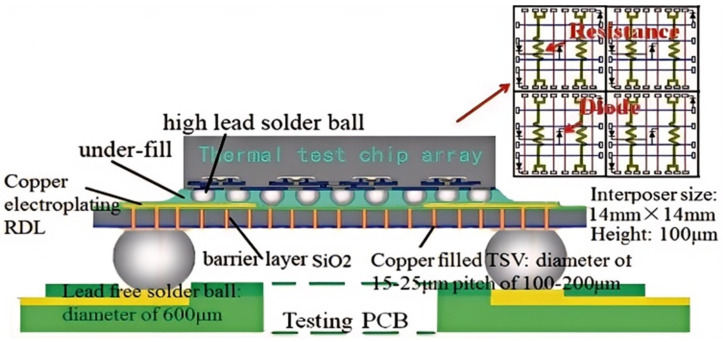
Schematic diagram of the structural design of the test tool. Reproduced with permission from [51]. Copyright 2012 IEEE.

**Figure 18 micromachines-16-00669-f018:**
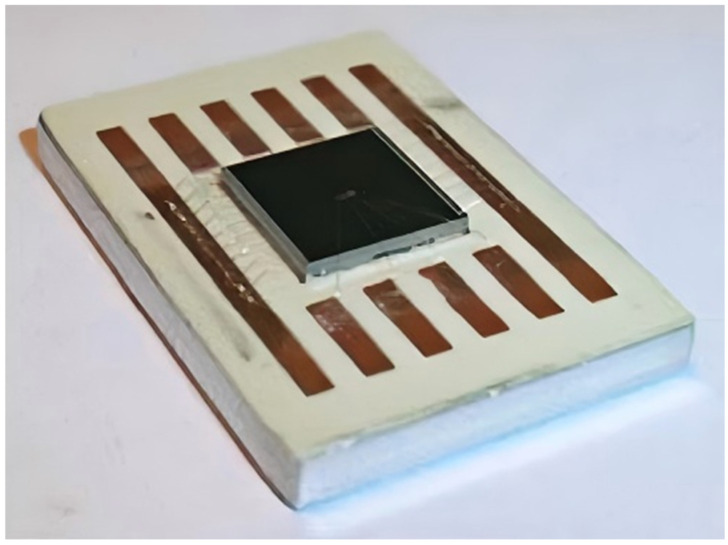
Structure of thermal conductivity test of dielectric layer. Reproduced with permission from [52]. Copyright 2006 IEEE.

**Figure 19 micromachines-16-00669-f019:**
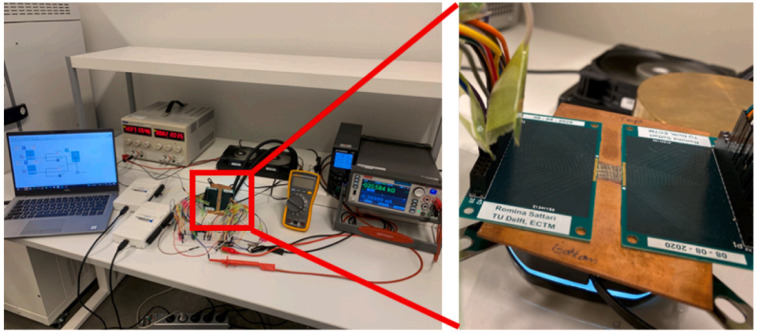
TDIM measuring device and bonded samples. Reproduced with permission from [45]. Copyright 2022 Elsevier Ltd.

**Figure 20 micromachines-16-00669-f020:**
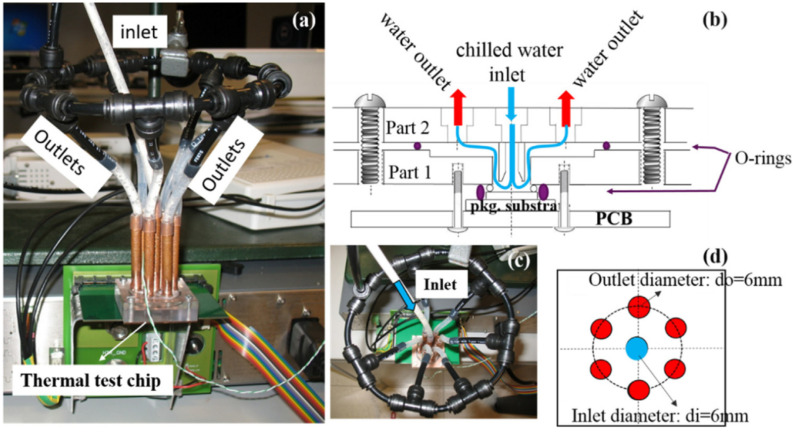
Demonstration of a chip-scale single-impact jet cooler: (**a**) experimental set-up photo of single jet cooler; (**b**) schematic view of the cooler with different parts; (**c**) and (**d**) photo and isometric drawing of single inlet and six outlets. Reproduced with permission from [38]. Copyright 2019 Elsevier Ltd.

**Figure 21 micromachines-16-00669-f021:**
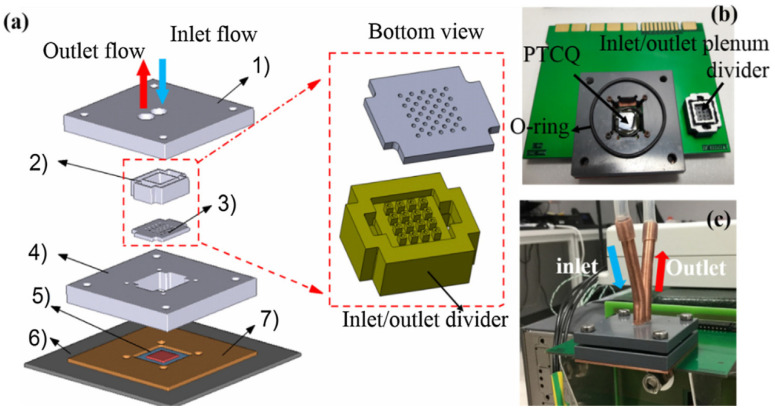
Demonstration of a chip-scale 4 × 4 shock-jet cooler: (**a**) CAD (computer aided design) design details of different individual parts; (**b**) photo with the arrangement of nozzle placement; (**c**) final assembly of 4 × 4 impingement jet cooler. (1-Cover layer, 2-inlet/outlet plenum, 3-nozzle plate, 4-support structure, 5-PTCQ (packaging test chip version Q) thermal test chip, 6-PCB, 7-copper spacer). Reproduced with permission from [38]. Copyright 2019 Elsevier Ltd.

**Figure 22 micromachines-16-00669-f022:**
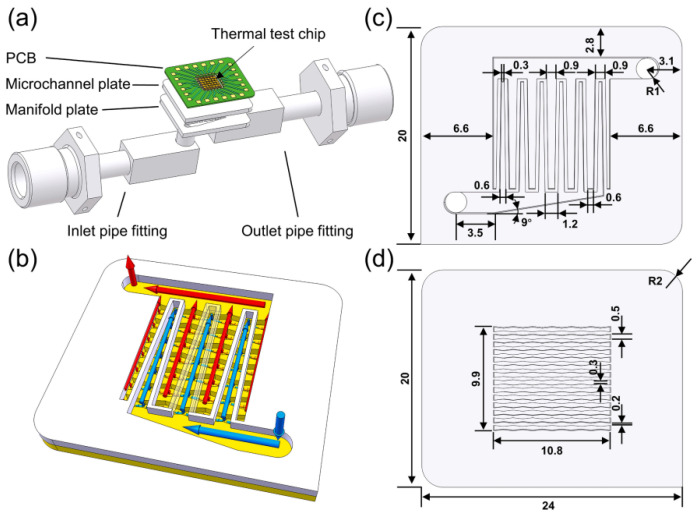
Mechanically pumped flow loop manifold microchannel radiator: (**a**) Assembly drawing of the test section; (**b**) schematic of MMCHS (manifold microchannel heat sink) (not to scale); structure and dimensions of (**c**) manifold plate and (**d**) microchannel plate (unit: mm). Reproduced with permission from [53]. Copyright 2024 Elsevier Ltd.

**Table 1 micromachines-16-00669-t001:** Thermal test chip parameters. Data from [29,30,37,39,43,46,47,48].

Chip Parameters	NT16-3k [46]	NT20-3k [47]	TEA [29]	IMB-CNM (CSIC) [30]	Xilinx [39]	IMEC [37]	Delft [43]	SMIC [48] *
**Heater type**	Resistive	Resistive	Resistive	Resistive	Resistive, RO	Resistive	Resistive	Resistive
**Sensor type**	RTD	RTD	Diode	RTD	RTD, RO, Diode	Diode	RTD	RTD
**Chip structure**	Unit array	Unit array	Unit array	Unit array	Unit array	Unit array	Unit array	Unit array
**Unit size (mm^2^)**	3.2 × 3.2	2.5 × 2.5	2.54 × 2.54	6 × 6	NA	0.24 × 0.24	4 × 4	25 × 16
**Number of heaters in unit**	10	2	2	130	NA	2	6	15
**Number of sensors in unit**	1	1	4	1	NA	1	3	7
**Percentage of heating area**	62.5%	82%	86%	98%	NA	75%	82.5%	NA
**Temperature range**	10 to 100 °C	NA	NA	20 to 200 °C	30 to 125 °C	10 to 75 °C	−55 to 150 °C	NA
**Temperature sensitivity**	8.2 Ω/K	10 Ω/K	−2 mV/ °C@1 mA	0.95 °C/Ω	±2 °C	−1.55 ± 0.02 mV/°C@5 μA	Linear RTD: 12 Ω/K Spiral RTD: 9 Ω/K	NA

* represents no actual fabrication, parameters are design values, power is simulation value.

**Table 2 micromachines-16-00669-t002:** Sample configurations. Adapted with permission from [49]. Copyright 2013 IEEE.

Configuration	Chip Bottom	Chip Top
1a	5.08 mm × 5.08 mm WB	NA
1b	5.08 mm × 5.08 mm WB	2.54 mm × 2.54 mm WB
2a	5.08 mm × 5.08 mm FC	NA
2b	5.08 mm × 5.08 mm FC	2.54 mm × 2.54 mm WB

**Table 3 micromachines-16-00669-t003:** Thermal test chip application scenarios. Data from [37,38,45,49,50,51,52,53].

Application Scenarios	Characterization of Chip Temperature Distribution			Integrated Process Testing			Chip Heat Dissipation Research	
**Specific application**	Stacked chip thermal distribution [49]	Heat reflection imaging [50]	Thermal distribution of 3D package structure [37]	Silicon through-hole performance testing [51]	In situ measurement of dielectric thermal conductivity [52]	Sintered structure shell thermal resistance measurement [45]	Liquid jet impact cooling heat dissipation characterization [38]	Manifold microchannel heat dissipation characterization [53]
**Chip types**	TEA (TTC-1002)		IMEC	TEA (TTC-1002)	IMB-CNM (CSIC)	Delft	IMEC	BUAA
**Heater type**	Resistive		Resistive	Resistive	Resistive	Resistive	Resistive	Resistive
**Sensor type**	Diode		Diode	Diode	RTD	RTD	Diode	Diode
**Chip structure**	Unit array		Unit array	Unit array	Unit array	Unit array	Unit array	Unit array
**Unit size (mm^2^)**	2.54 × 2.54		0.24 × 0.24	2.54 × 2.54	6 × 6	4 × 4	0.24 × 0.24	1 × 1
**Number of heaters in unit**	2		2	2	130	6	2	NA
**Number of sensors in unit**	4		1	4	1	3	1	1
**Percentage of heating area**	86%		75%	86%	98%	82.5%	75%	>85%
**Temperature range**	NA		10 to 75 °C	NA	20 to 200 °C	−55 to 150 °C	10 to 75 °C	NA
**Temperature sensitivity**	−2 mV/ °C@1 mA		−1.55 ± 0.02 mV/°C@5 μA	−2 mV/ °C@1 mA	0.95 °C/Ω	Linear RTD: 12 Ω/K Spiral RTD: 9 Ω/K	−1.55 ± 0.02 mV/°C@5 μA	NA

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
