# Peer review of "Overview of Research Progress and Application Prospects of Thermal Test Chips"

_micromachines, 2025, doi:10.3390/mi16060669_

Round 1
Reviewer 1 Report
Comments and Suggestions for Authors
This is a review paper covering the development of thermal test chips, from design, simulation, manufacturing and characterization, spanning more than two decades of research.
Overall the paper is well written, and the analysis is worthy, but there are some issues that have to be addressed by the authors before the paper can be accepted. The most important are listed herein.
a) The writing is not uniform in some aspect. After a period, there should be a space before the next sentence begins. This manuscript does that partially, but in many occasions it just begins the sentence right after the period. Please include the space in all instances.
b) All acronyms should be defined the first time they are used, and only then. The manuscript presents many instances where a previously defined acronym is defined again, as well as many cases where the acronym is never defined.
c) PMOS is not “Positive MOS”; it is P-channel MOS.
d) When defining area, the authors generally indicate the units in each factor, which is adequate, but this form should be used throughout the manuscript (for instance, on line 583 the units of the first factor are missing). Moreover, in some instances the authors include a space between the number and the units, in most they don’t. Please be consistent.
e) Tables 1 and 2 are cut on the right side. Please scale them so that they show in their entirety.
f) There are several typing mistakes. Some are the following. On line 25 analysis should read analyses, since the plural form must be used according to the context. On line 337, pleas include a space after the Celsius symbol (C). Please rewrite lines 361-362 in order to avoid the repeated use of “standard”. On 573, it should read “a demand”.
Comments on the Quality of English LanguageThere are a few instances where the language has to be corrected, please see my comments above.
Reviewer 2 Report
Comments and Suggestions for Authors
In this article, Lina Ju et al. present a review of thermal test chip (TTC) technology as a key solution for managing thermal and stress challenges in advanced semiconductor packaging. They highlight the limitations of traditional simulation methods and emphasize TTC’s role in in-situ testing during packaging and heat dissipation design. The study summarizes global research progress and discusses the need to evolve TTCs from single-parameter to multi-physical field sensors, integrating temperature, stress, electromigration, and electromagnetic measurements, to meet the demands of increasingly complex microsystems.
In this review article, the authors effectively cover the evolution of TTC research from the start to the current developments, which truly makes the article highly suitable for researchers working in this field. The manuscript is well organized and clearly written in a very systematic way, and is easy to follow. Based on the content and presentation, I believe the current version of the manuscript is suitable for publication.
However, there are a few minor issues in the manuscript that should be addressed before publication. For example, #Please ensure that numerical values are properly spaced from their units throughout the manuscript, # Some tables appear misaligned or partially cut off in the PDF version. This may be due to issues during the Word-to-PDF conversion. It is recommended to check the formatting and ensure all tables are clearly visible and properly aligned. # Minor typos and formatting inconsistencies need to be corrected to enhance the readability of the manuscript.
